# Geometric Optimization of Perovskite Solar Cells with Metal Oxide Charge Transport Layers

**DOI:** 10.3390/nano12152692

**Published:** 2022-08-05

**Authors:** Jasurbek Gulomov, Oussama Accouche, Rayimjon Aliev, Bilel Neji, Raymond Ghandour, Irodakhon Gulomova, Marc Azab

**Affiliations:** 1Renewable Energy Sources Laboratory, Andijan State University, Andijan 170100, Uzbekistan; 2Andijan State Pedagogical Institute, Andijan 170100, Uzbekistan; 3College of Engineering and Technology, American University of the Middle East, Egaila 54200, Kuwait

**Keywords:** photovoltaics, perovskite, metal oxide, solar cell, Sentaurus TCAD, photoelectric parameters, electron transport layer, hole transport layer

## Abstract

Perovskite solar cells (PSCs) are a promising area of research among different new generations of photovoltaic technologies. Their manufacturing costs make them appealing in the PV industry compared to their alternatives. Although PSCs offer high efficiency in thin layers, they are still in the development phase. Hence, optimizing the thickness of each of their layers is a challenging research area. In this paper, we investigate the effect of the thickness of each layer on the photoelectric parameters of *n-ZnO/p-CH_3_NH_3_PbI_3_/p-NiO_x_* solar cell through various simulations. Using the Sol–Gel method, PSC structure can be formed in different thicknesses. Our aim is to identify a functional connection between those thicknesses and the optimum open-circuit voltage and short-circuit current. Simulation results show that the maximum efficiency is obtained using a perovskite layer thickness of 200 nm, an electronic transport layer (ETL) thickness of 60 nm, and a hole transport layer (HTL) thickness of 20 nm. Furthermore, the output power, fill factor, open-circuit voltage, and short-circuit current of this structure are 18.9 mW/cm^2^, 76.94%, 1.188 V, and 20.677 mA/cm^2^, respectively. The maximum open-circuit voltage achieved by a solar cell with perovskite, ETL and HTL layer thicknesses of (200 nm, 60 nm, and 60 nm) is 1.2 V. On the other hand, solar cells with the following thicknesses, 800 nm, 80 nm, and 40 nm, and 600 nm, 80 nm, and 80 nm, achieved a maximum short-circuit current density of 21.46 mA/cm^2^ and a fill factor of 83.35%. As a result, the maximum value of each of the photoelectric parameters is found in structures of different thicknesses. These encouraging results are another step further in the design and manufacturing journey of PSCs as a promising alternative to silicon PV.

## 1. Introduction

The race towards a completely sustainable, green, and zero-emission electricity has led to many challenges on various levels, especially at the end of energy production. While hydropower production is still leading the way as primary renewable-energy resource, solar-energy production is showing a strong and an exponential growth, from 30 TWh to more than 1000 TWh, during the last 10 years, according to our world in data [1]. This boost is steered with political supports and tax initiatives in many countries. Nowadays, numerous ongoing researches are being conducted to develop sustainable solar cells that are inexpensive, efficient, and with scalable production. On the industrial level, 95% of solar cells are made of silicon [2], offering a maximum efficiency of 29% [3], with costly processes when compared to the production processes of non-silicon-based solar cells [4]. This resulted in the exploration of new easy-to-synthesize materials to create sustainable and highly efficient solar cells by researchers [5].

The first perovskite solar cell was invented in 2009 with an efficiency of 3.8% [6]. In fact, the perovskite material can be synthesized using the Sol–Gel method, which is a straightforward process that can be conducted inside a simple laboratory at a reasonable cost. Consequently, the interest in perovskite and the number of related scientific works have increased significantly [7]. Perovskite materials are divided into two types: organic–inorganic hybrids and inorganic [8]. Today, lead–halide organic–inorganic perovskite materials are widely used in solar cells. The maximum measured efficiency of a lead–halide perovskite solar cell is 25.8% [9] while the theoretical efficiency is equal to 31% according to Shockley–Queisser theory [10]. In perovskite materials, the maximum conversion efficiency can be achieved with very thin layers due to its high light absorption coefficient. However, the toxicity of the materials, device hysteresis, and instability are its main disadvantages. Such toxicity comes from the presence of Pb+ ions [11] in its materials. Therefore, it is recommended to use lead-free perovskite materials to reduce the toxicity of solar cells. On the other hand, the instability of perovskite solar cells is due to the hydroscopic properties of these organic cations, especially methylammonium. The stability of CH_3_NH_3_PbI_3_-based solar cells is mainly affected by moisture [12]. This is because water molecules form weak hydrogen bonds with perovskite cations, which disrupt the structural stability of the crystal [13]. Moreover, other halogen elements are used instead of iodine to increase the stability of perovskite solar cells [14], as the evaporation of the iodine decreases the stability of the CH_3_NH_3_PbI_3_ solar cell.

Simple perovskite-based solar cells consist of an electron transport layer (ETL), a light-absorbing perovskite layer, and a hole transport layer (HTL). TiO_2_, SnO_2_, In_2_S_3_, and ZnO materials are commonly used as ETL [15] and Cu_2_O, CuSCN, NiO_x_, and Spiro-Ometad materials are used as HTL [16] in CH_3_NH_3_PbI_3_-based solar cells. Perovskite solar cells absorb light mostly in the ultraviolet and visible spectrums. Accordingly, tandem solar cells with high efficiency are formed with silicon-based solar cells [17]. According to the Shockley–Queisser limit, the maximum efficiency of silicon/perovskite tandem solar cells can reach 45% [18], whereas the efficiency of a multijunction tandem solar cell can even exceed 50% [19]. To improve the optical properties of a solar cell, and, hence, increase its efficiency, texturing method [20] and metal nanoparticles [21] are employed.

When NiO_x_ is used as the HTL layer, the output power of the CH_3_NH_3_PbI_3_-based solar cell stays unchanged for up to 4000 h [22], thus we decided to employ NiO_x_ as an HTL layer in our study. Besides, ZnO was used as an ETL layer for three main reasons. Firstly, ZnO is transparent; secondly, it acts like an antireflection coating; and, thirdly, it forms a charge transport layer for CH_3_NH_3_PbI_3_-based solar cells. Although, using ZnO in solar cell applications has its own drawbacks, mainly its degradation under UV light. This well-known problem was discussed properly and solved in Mario Prosa’s work [23]. It was found with theoretical proof that the ultraviolet stability of solar cells is improved when using Al doped ZnO. Accordingly, Al doped ZnO is used in this work. Moreover, NiO_x_ and ZnO can be synthesized experimentally using the Sol–Gel method, where forming *p-type* and *n-type* is a straightforward process. Figure 1 shows the structure (Figure 1a) and band structure (Figure 1b) of the *n-ZnO/p-CH_3_NH_3_PbI_3_/p-NiO_x_* solar cell studied in this paper. According to the band structure, ZnO and NiO_x_ are the most suitable materials for the ETL and HTL layers of the CH_3_NH_3_PbI_3_-based solar cell. Perovskite solar cells are more sensitive to geometric dimensions than traditional silicon-based solar cells. Therefore, one of the important tasks is to find the optimal thickness of each layer. Such thicknesses can be controlled during the manufacturing process using the Sol–Gel method. Determining the optimal thicknesses of each layer of perovskite solar cells will allow the creation of an efficient solar cell while leaving the door wide open for researchers to explore the effect of metal nanoparticles or textures on solar cells.

This paper is organized as follows: Section 1 presents the introduction. The used method is presented in the Section 2. Simulations and results are discussed in Section 3, to end with a conclusion and future works.

## 2. Materials and Methods

### 2.1. Simulation

There are three main methods for studying solar cells: experiment, theory, and simulation. In this scientific work, the simulation method is applied. Technology Computing Aided Design (TCAD) software is widely employed in the simulation of semiconductor devices, in addition to analyzing product quality before setting up a production process. Silvaco TCAD, Sentaurus TCAD [24], Lumerical TCAD [25], and Comsol Multiphysics [26] are well-known for modeling solar cells. In this study we are using Sentaurus TCAD to simulate perovskite solar cells. Sentaurus TCAD has many tools, and four of these are used to simulate solar cells [27]: Sentaurus Structure Editor, Sentaurus Device, Sentaurus Workbench, and Sentaurus Visual.

The geometric model of the solar cell is created using Sentaurus Structure Editor. In our previous research, we covered in detail how to create a geometric model of solar cells by developing an algorithm using the Tool Command Language (TCL) in the Sentaurus Structure Editor [28]. The aim of this work is to analyze the impact of varying the light-absorbing perovskite layer of the solar cell, between 200 and 800 nm, and to study the impact of varying the thickness of the Electron Transport Layer (ETL) and Hole Transport Layer (HTL) between 20 nm to 100 nm on the performance of solar cells. Accordingly, n-type ZnO was concluded as ETL, whereas p-type NiO_x_ as HTL. Additionally, Al atoms with 1 × 10^17^ cm^−3^ concentration were introduced into the ETL layer, forming n-type ZnO, due to the fact that Zn and O are divalent elements, whereas Al is a trivalent element, therefore, introducing Al atoms into ZnO results in forming an n-type conduction. NiO_x_ is a p-type due to the vacancy of Ni. P type is formed by doping it with Sn. However, in this scientific study, a vacancy is used to form a p-type in NiO_x_ since vacancies are responsible for forming p-type junctions during the synthesis process of NiOx, using the Sol–Gel method.

### 2.2. Physical Parameters of Materials

After creating the geometric model of our solar cell, the physical parameters are assigned to the materials. The Sentaurus TCAD database includes parameter files of more than 40 materials [29], among which the properties of ZnO, NiO_x_, and CH_3_NH_3_PbI_3_ are missing. The unavailable materials properties can be created and added to the database manually. Therefore, the physical properties and parameters of ZnO, CH_3_NH_3_PbI_3_, and NiO_x_ were collected from literature and expressed in Table 1.

The optical parameters of materials depend on the wavelength of incident light. Figure 2 shows the dependence of real (a) and imaginary (b) parts of the complex refractive index of ZnO [41], NiO_x_ [42], and CH_3_NH_3_PbI_3_ [43] on the light wavelength. The properties of materials were chosen according to their synthesis method. Accordingly, we employed materials that are synthesizable using the Sol–Gel method.

### 2.3. Physical Background of Simulation

The simulation of the solar cell is performed in two stages. The optical properties are determined during the first stage while the electrical determination is performed in the second stage. The Sentaurus TCAD has a Transfer Matrix Method (TMM), Beam Propagating Method (BPM), and Ray Tracing Method to determine the optical properties. In this study, the TMM given in Equation (1) was used to determine the optical properties of solar cells as it considers the phenomenon of interference in thin layers [44]. Consequently, this method is the most suitable option for studying the effect of layer thickness on the properties of the solar cell. TMM can be used to model the effect of light trapping due to scattering of incident light rough interface of a planar multilayer structure. Part of incident light at a rough interface is scattered and spread incoherently. In Sentaurus Device, the TMM method uses Scalar Scattering Theory to approximate the amount of scattered light at a rough interface. Besides, directional dependency of the scattering process is modeled by Angular Distribution functions (ADF). The so-called haze parameter defines the ratio of diffused (scattered) light to total (diffused + specular) light. In this scenario, a rough interface is characterized by its haze function (haze parameter as a function of the wavelength of incident light) for reflection and transmission as well as the ADF for the direct coherent light incident at the interface and the ADF for the scattered light incident at the interface.
(1)$$\left[\begin{array}{l} E_{i}\\ E_{r} \end{array}\right]=M\left[\begin{array}{l} E_{t}\\ 0 \end{array}\right]$$
where, *M* is the matrix, *E_i_* is the electrical field of the incident light, *E_r_* is the electric field of the reflected light, and *E_t_* is the electric field of the transmitted light.

In stage two, the Poisson equation, the Fermi function, and the continuity equations are used to determine the electrical properties. By calculating the Poisson equation given in Equation (2), the electric field strength and potential at different points of the solar cell are determined.
(2)$$\Delta \varphi =-\frac{q}{\varepsilon }\left(p-n-N_{D}+N_{A}\right)$$
with *ε* is the permittivity, *n* and *p* are the electron and hole concentrations, *N_D_* and *N_A_* are the concentrations of donor and acceptor, and *q* is the charge.

Using the Fermi function given in Equation (3), the concentration of charge carriers at different points in the solar cell is calculated.
(3)$$n=N_{c}F_{1/2}\left(\frac{E_{F,n}-E_{c}}{kT}\right)\;\;\;\;and\;\;\;\;p=N_{V}F_{1/2}\left(\frac{E_{V}-E_{F,p}}{kT}\right)$$
where *N_c_* and *N_v_* are the densities of states in conduction and valence bands, *E_c_* is the minimum energy of conduction band, *E_v_* is the maximum energy of valence band, *T* is the temperature, *k* is the Boltzmann constant, and *E_F,n_* and *E_F,p_* are the quasi-fermi energies.

The continuity equation given in Equation (4) is used to calculate the current due to the displacement of the charge carriers. In order to solve the Sentaurus TCAD continuity equation, four different models can be used: Drift–Diffusion, Thermodynamics, Hydrodynamics, and Monte Carlo. Both, thermodynamics and hydrodynamics models consider the effect of temperature on the charge transport. In contrast, the Monte Carlo method works by using statistical analysis to determine the probability of different outcomes with the presence of some randomness. In this work, the Drift–Diffusion model is adopted to calculate the carrier transport, since the effect of temperature on the performance of the solar cell was not considered.
(4)$$\begin{array}{l} \nabla \cdot {\vec{J}}_{n}=qR_{net,n}+q\frac{\partial n}{\partial t}\\ -\nabla {\vec{J}}_{p}=qR_{net,p}+q\frac{\partial p}{\partial t} \end{array}$$
where: *J_n_* and *J_p_* are the current densities of electron and holes, *R_net,p_* and *R_net,n_* are the net recombination of electron and holes, and *t* is the time.

An Ohmic contact is made to transmit electricity from the solar cell to the network because its resistance is too small. For that reason, the electrical boundaries conditions were calculated using the Ohmic boundary conditions given in Equation (5):(5)$$\begin{array}{l} \varphi =\varphi_{F}+\frac{kT}{q}a\sinh\left(\frac{N_{D}-N_{A}}{2n_{i,eff}}\right)\;\\ n_{0}p_{0}=n_{i,eff}^{2}\\ n_{0}=\sqrt{\frac{{\left(N_{D}-N_{A}\right)}^{2}}{4}+n_{i,eff}^{2}}+\frac{N_{D}-N_{A}}{2}\\ \;p_{0}=\sqrt{\frac{{\left(N_{D}-N_{A}\right)}^{2}}{4}+n_{i,eff}^{2}}-\frac{N_{D}-N_{A}}{2} \end{array}$$
Here: *n_i,eff_* is the effective intrinsic carrier concentration, and *φ_F_* is the Fermi potential of contact.

Finally, since we are employing a numerical approach, the solar cell is meshed with a mesh size varying from 2 nm to 4 nm.

## 3. Results and Discussion

The main relevant parameters of solar cells are the fill factor, the open-circuit voltage, the short-circuit current, and efficiency. These parameters are investigated in the next subsections.

### 3.1. Open-Circuit Voltage

The open-circuit voltage depends on the size of the solar cell. This is because a change in the size of the solar cell affects the carrier concentration, which leads to a different open-circuit voltage. Figure 3 describes the dependence of the open-circuit voltage of *n-ZnO/p-CH_3_NH_3_PbI_3_/p-NiO_x_* solar cell with a perovskite layer of 200 nm (a), 400 nm (b), 600 nm (c), and 800 nm (d) thicknesses on ETL and HTL layer thicknesses. Contour graphs show the dependence of the open-circuit voltage on the thickness of the ETL and HTL layers simultaneously. They are three-dimensional graphs which can help in the analysis and the optimization of solar cells. There are many optimal values for the thicknesses of ETL and HTL layers at which solar cells can reach their maximum open-circuit voltage. If the results in Figure 3a are called matrix elements, depending on the thicknesses of ETL and HTL, then when the thicknesses of the ETL and HTL layers are equal to the indices on the diagonal of this matrix, the solar cell reaches its maximum open-circuit voltage. At 200 nm of thickness of a perovskite layer, the solar cell with ETL and HTL layer thicknesses, satisfying the condition in Equation (6), reaches a maximum open-circuit voltage.
(6)$$d_{etl}+d_{htl}=120\;nm,\;d_{htl}\leq 100\;nm,\;d_{etl}\leq 100\;nm$$
with: *d_etl_* is the thickness of the ETL layer, and *d_htl_* is the thickness of the HTL layer.

As the thickness of the perovskite layer increases, the relationship between the values of the optimal thickness of ETL and HTL violates the conditions of Equation (6). In general, it was found that the optimal value of ETL and HTL for the open-circuit voltage changes when the thickness of the perovskite layer changes. Additionally, in the range of ETL and HTL layer thicknesses of 20–80 nm and perovskite layer thickness of 200–800 nm, the optimal thicknesses to achieve the maximum open-circuit voltage can be determined using the condition given in Equation (7). This condition was formed based on the results obtained and presented in Figure 3. Figure 3c shows that the solar cell with perovskite layer thickness of 600 nm (*d_p_* = 600 nm), ETL and HTL layer thicknesses of 40 nm (*d_etl_* = 40 nm and *d_htl_* = 40 nm) reaches the maximum open-circuit voltage, while it keeps satisfying the condition in Equation (7).
(7)$$\{\begin{array}{c} d_{etl}+d_{htl}=140-\frac{d_{p}}{10}\;nm,\text{ if 20 nm}\leq d_{htl},d_{etl}\leq 120-\frac{d_{p}}{10}\;nm\\ d_{etl}+d_{htl}=220-\frac{d_{p}}{10}\;nm,\text{ if }120-\frac{d_{p}}{10}\;nm\leq d_{htl},d_{etl}\leq 100\;nm \end{array}$$
here: *d_p_* is the perovskite layer thickness.

Changes in the thickness of the ETL and HTL layers affect the internal electric field. As the electric potential difference between the contacts (anode and cathode) changes the open-circuit voltage changes. The open-circuit voltage of the solar cell is directly related to the band gap energy of the perovskite material and is inversely related to the intrinsic carrier concentration. Likewise, the intrinsic carrier concentration is also inversely related to band gap energy. According to [42], the band gap energy of CH_3_NH_3_PbI_3_ is equal to 1.56 eV and its intrinsic carrier concentration to n_i_ = 8 × 10^4^ cm^−3^ [45]. Hence, the open-circuit voltage of the *n-ZnO/p-CH_3_NH_3_PbI_3_/p-NiO_x_* solar cell is greater than 1.176 V, which is the minimum value of the open-circuit voltage. Moreover, parasitic light absorption has a negative effect on the open-circuit voltage [46]. The amount of parasitic absorption strongly depends on the thickness of each layer of the solar cell. Parasitic absorption occurs in the ETL layer due to the high surface recombination rate [47]. Accordingly, at a thickness of 800 nm of the perovskite layer, the open-circuit voltage decreases with an increasing ETL layer thickness.

### 3.2. Short-Circuit Current

The short-circuit current of a solar cell depends directly on the thickness of the layers. This is because the thickness of the layers affects the concentration of absorbed photons. Figure 4 depicts the dependence of short-circuit current of the *n-ZnO/p-CH_3_NH_3_PbI_3_/p-NiO_x_* solar cell with a thickness of 200 nm (a), 400 nm (b), 600 nm (c), and 800 nm (d) of the perovskite layer on the ETL layer and the HTL layer thicknesses. According to the results given in Figure 4, it was found that the thickness of the HTL layer should be less than 60 nm in order to maximize the short-circuit current. The optimal thickness of the ETL layer, on the other hand, varied depending on the thickness of the perovskite layer for a maximum value of short-circuit current. As shown in Figure 4a, the ETL layer thickness should be in the range of 40 *≤ d_etl_ ≤* 80 nm and its value should satisfy the condition *d_etl_ + d_htl_* = 100 nm.

Figure 4b,c showed that the ETL and HTL layer thicknesses should respect the conditions of Equation (8), to maximize the short-circuit current of the solar cell when the perovskite layer thickness is between 400 nm and 600 nm. In addition, when the thickness of the perovskite layer is 600 nm, the thickness of the ETL layer should be between 80–100 nm and the thickness of the HTL layer should range between 20–60 nm to achieve the maximum short-circuit current.
(8)$$d_{etl}+d_{htl}\leq 120-\frac{d_{p}}{10}\;nm,\;\mathrm{if}\;20\;\mathrm{nm}\leq d_{htl},d_{etl}\leq 100-\frac{d_{p}}{10}\;nm$$

According to Figure 4d, when the thickness of the perovskite layer is 800 nm, the ETL and HTL layer thicknesses should be selected according to the condition given in Equation (9) to maximize the short-circuit current.
(9)$$80\;nm\leq d_{etl}+d_{htl}\leq 120\;nm,\;\mathrm{if}\;\{\begin{array}{c} 20\;\mathrm{nm}\leq d_{htl}\leq 60\;nm\\ 60\;\mathrm{nm}\leq d_{etl}\leq 100\;nm \end{array}\;$$

When Perovskite, ETL and HTL layer thicknesses of solar cell equal to (200 nm, 100 nm, and 20 nm), (400 nm, 80 nm, and 20 nm), (600 nm, 60 nm, and 20 nm), and (800 nm, 40 nm, and 20 nm), respectively, short-circuit current was found to be minimum. In general, the maximum value of the short-circuit current increased as the thickness of the perovskite layer increased. This is due to the increase in the thickness of the perovskite layer, which maximizes the probability of absorption of photons in the infrared field and improves the quantum efficiency of the solar cell in this field at the same time [48]. However, changes in the ETL and HTL layers cause changes in the concentration of photons absorbed in the perovskite layer. The ETL layer also acts as an anti-reflective layer for the perovskite solar cell. Accordingly, changing the thickness of ETL causes a variation in the interference phenomenon [49] and, therefore, a variation of the reflection coefficient. This can be explained by the direct correlation between the short-circuit current and the concentration of the absorbed photons.

### 3.3. Fill Factor

One of the most important parameters of the solar cell is the fill factor. It can be used to estimate the series and parallel resistances of a cell. Figure 5 presents the dependence of fill factor of *n-ZnO/p-CH_3_NH_3_PbI_3_/p-NiO_x_* solar cell with perovskite layer thickness of 200 nm (a), 400 nm (b), 600 nm (c), and 800 nm (d) on ETL and HTL layer thicknesses. It can be observed that the maximum value of the fill factor did not follow the same variation functions obtained in the case of the voltage and short-circuit current.

The maximum fill factor of 83.35% could be reached when the perovskite, ETL and HTL layer thicknesses of a solar cell are equal to (200 nm, 100 nm, and 40 nm), (400 nm, 80 nm, and 40 nm), (600 nm, 80 nm, and 80 nm), and (800 nm, 40 nm, and 40 nm), respectively. The fill factor of the solar cell at the thicknesses of (600 nm, 80 nm, and 80 nm) is 83.35%. It is highest value of fill factor. However, in practice [50], the maximum value of the fill factor depends mainly on the series resistance, shunt resistance, and recombination rate. For an ideal solar cell, that has an ideality coefficient (*n* = 1), can be expressed in the below equation [51]:
(10)$$FF=\frac{V_{oc}-\ln\left(V_{oc}+0.72\right)}{V_{oc}+1}$$
with: *V_oc_* is the open-circuit voltage, and *FF* is the fill factor.

An ideal fill factor for a solar cell with an open-circuit voltage greater than 1 V is around 90%. As the ideal coefficient increases, the fill factor decreases. Figure 6 shows the time dependence of the Shockley–Read–Hall (SRH) and Auger recombination rates in a solar cell of thicknesses 200 nm, 60 nm, and 20 nm. Hence, the SRH recombination rate in this structure is 1000 times greater than that of Auger. Therefore, the ideal coefficient of this solar cell can be two (*n* = 2). Hence, the corresponding fill factor is 83.35%. According to Nandi Wu’s experimental results [52], the fill factor of the perovskite solar cell is 83% when the ideal coefficient is two (*n* = 2). Consequently, the results of our simulation modeling are consistent with the experimental results.

### 3.4. Output Power

Figure 7 describes the dependence of fill factor of *n-ZnO/p-CH_3_NH_3_PbI_3_/p-NiO_x_* solar cell with perovskite layer thickness of 200 nm (a), 400 nm (b), 600 nm (c), and 800 nm (d) on ETL and HTL layer thicknesses. The maximum output power was recorded in perovskite solar cells with ETL and HTL layer thicknesses of (200 nm, 60 nm, and 20 nm), (400 nm, 40 nm, and 40 nm), (600 nm, 20 nm, and 40 nm), and (800 nm, 80 nm, and 40 nm). Among the different formed structures, a solar cell with thicknesses of 600 nm, 20 nm, and 40 nm resulted in a maximum output power of 18.90 mW/cm^2^. The range of optimal values of HTL layer thickness did not change with increasing perovskite layer thickness. However, the optimal thickness of the ETL has changed. Results of Figure 7a show that the optimal ETL and HTL layer thicknesses for a solar cell with a perovskite layer thickness of 200 nm was in the range of 40–80 nm, and 20–40 nm, respectively. At a perovskite layer thickness of 400 nm, the optimal value was in the range of 20–40 nm for HTL, and in the range of 20–60 nm for ETL. It could be concluded that an increase of 200 nm in the perovskite layer thickness results in a 20 nm decrease in the maximum and minimum values of the optimal value range of the ETL layer. However, according to the results of Figure 7d, when the perovskite layer thickness increased by 600 nm, the limits of the optimal value range of the ETL layer thickness increased by 20 nm.

Both the best and worst solar cells were observed when the perovskite layer was 200 nm thick. The output power of a solar cell (200 nm, 20 nm, and 20 nm) was at a minimum of 13.84 mW/cm^2^; therefore, a solar cell with a structure of this size has the worst performance. The high output power of a perovskite solar cell in thin layers is due to the high absorption coefficient. The only difference of solar cells with high and low output power lies within the thickness of the ETL layer. As ETL acts like an anti-reflection layer, the change in its thickness mainly affects the concentration of photons absorbed in the perovskite layer. Figure 8, shows the dependance of the absorption (Figure 8a) and reflection (Figure 8b) coefficients of solar cells with sizes of (200 nm, 20 nm, and 20 nm) (1) and (200 nm, 60 nm, and 20 nm) (2) on light wavelength. In the light wavelength range of 445 to 800 nm, the absorption coefficient of the solar cell (200 nm, 60 nm, and 20 nm) was found to be higher. The increase in ETL layer thickness resulted in better absorption in the near-infrared region. Absorption in the long-wavelength range can also be improved by increasing the thickness of the perovskite layer. Nonetheless, the recombination rate also increases as the thickness of the perovskite layer increases. In fact, an increase in the thickness of the perovskite layer mainly reduces the transmission coefficient and, therefore, leads to an increase in the absorption coefficient. On the other hand, a change in the thickness of the ETL layer reduces the reflection coefficient and, therefore, increases the absorption coefficient. As shown in Figure 8b, increasing the thickness of the ETL layer from 20 nm to 60 nm results in a significant reduction in the reflection coefficient. Hence, the optimal ETL and HTL layer thicknesses for the *n-ZnO/p-CH_3_NH_3_PbI_3_/p-NiO_x_* solar cell are 60 nm and 20 nm, respectively.

The dependence of output power on the thickness of the perovskite layer without changing the optimal thicknesses of ETL and HTL was studied. Figure 9 presents the dependence of output power of solar cell with ETL and HTL layer thicknesses of (60 nm, and 20 nm) and (20 nm, and 20 nm) on perovskite layer thickness. The output power of solar cells mentioned in this study was determined under outdoor illumination where the intensity is one sun. This very same output power will decrease drastically under indoor illumination due to linear decreasing of short circuit current and non-linear decreasing of open-circuit voltage [53]. As mentioned in [54], the spectrum of an indoor illumination is narrower and the light intensity is shorter than that of an outdoor illumination. Dependence between photoelectric parameters of solar cells and outdoor/indoor illumination was studied in [55].

Accordingly, the maximum output power was observed in solar cells with ETL and HTL layer thicknesses of (60 nm, and 20 nm), while the minimum output power in those with thicknesses of (20 nm, and 20 nm). In the first case, the output power of the solar cell decreased, while in the second case it increased when increasing the thickness of perovskite layer. As the thickness of the perovskite layer increases, recombination rate increases along with the absorption coefficient and therefore a dysfunction in the dependence of output power on the thickness takes place. At a perovskite layer thickness of 600 nm a sharp dropping of the output power occurred due to the minimum interference in perovskite layer.

## 4. Conclusions

In the production of perovskite solar cells, the main focus is their cost and efficiency. Therefore, we studied the *n-ZnO/p-CH_3_NH_3_PbI_3_/p-NiO_x_* solar cell, which has a straightforward and inexpensive process of production, using Sentaurus TCAD simulation. The effects of the thicknesses of layers on the photoelectric parameters of the above structure were studied. Based on the results, tuning the thicknesses of perovskite, ETL, and HTL layer leads to significant gain in the power output performance. Therefore, the maximum open-circuit voltage and short-circuit current as well as the causal equation were identified. The optimum structures, where each photoelectric parameter reaches its maximum value, were determined. Because, metal oxides and perovskite-based structures can be used in optoelectronic devices, such as pyranometers, optical sensors, and solar cells. The optimal structure is application-dependent. It is chosen according to the type of the device and its application. For instance, the output power is the most important parameter for the solar cell while short-circuit current is much more important for pyranometer applications. When perovskite layer thickness falls within the range of 200 nm to 800 nm, and when ETL and HTL layer thickness belongs to the interval of 20 nm to 100 nm, it was found that solar cells with thicknesses of (200 nm, 60 nm, and 20 nm) delivered the maximum output power while the minimum power was found in solar cell with thicknesses of (200 nm, 20 nm, and 20 nm). The maximum and minimum output power of the previous structures was found to be 18.9 mW/cm^2^ and 13.85 mW/cm^2^, respectively. In addition, the results obtained from our simulation were consistent with the experimental results. In summary, it is not necessary to continuously increase the thickness of the perovskite layer to achieve the maximum efficiency of the perovskite-based solar cell, indeed, it can be achieved by finding the optimal values of the ETL and HTL layer thicknesses., which is preferable because increasing the thickness of the perovskite layer will not only increase the absorption coefficient, but also increase the recombination rate. We successfully determined the optimal values of thicknesses of each layer of perovskite-based solar cells. These results may not help to sufficiently improve the efficiency of perovskite solar cells, but by creating nano-textures on the surface of perovskite-based solar cells with optimal thicknesses or introducing various metal nanoparticles into the ETL layer, the absorption coefficient and efficiency can be further increased. In our further scientific research, we will determine the optimal sizes of nanoparticles introduced in this structure and nano-textures formed on its surface.

## Figures and Tables

**Figure 1 nanomaterials-12-02692-f001:**
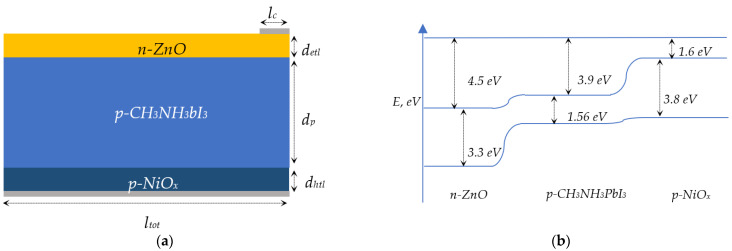
Geometric model (**a**) and band structure (**b**) of *n-ZnO/p-CH_3_NH_3_PbI_3_/p-NiO_x_* solar cell.

**Figure 2 nanomaterials-12-02692-f002:**
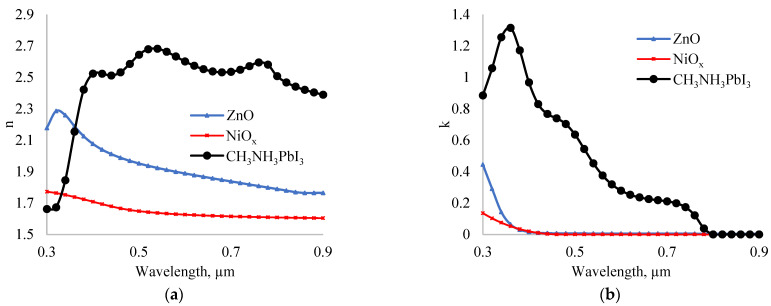
Dependence of real (**a**) and imaginary (**b**) parts of the complex refractive index of ZnO, NiO_x_, and CH_3_NH_3_PbI_3_ on the incident light wavelength.

**Figure 3 nanomaterials-12-02692-f003:**
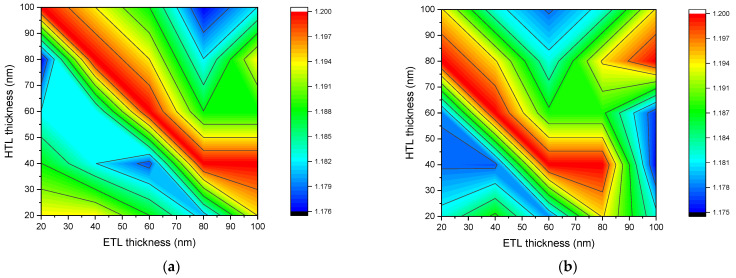
Dependence of open-circuit voltage of *n-ZnO/p-CH_3_NH_3_PbI_3_/p-NiO_x_* solar cell with perovskite layer thickness of 200 nm (**a**), 400 nm (**b**), 600 nm (**c**), and 800 nm (**d**) on the thicknesses of the ETL and the HTL layers.

**Figure 4 nanomaterials-12-02692-f004:**
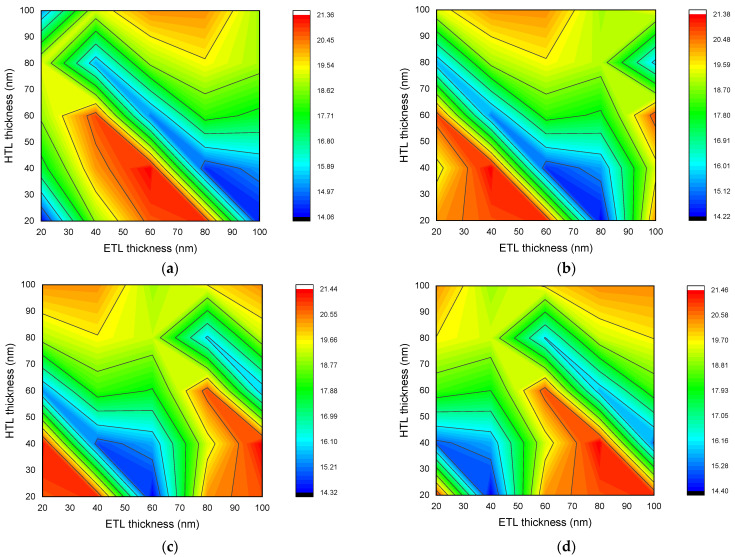
Dependence of short-circuit current of *n-ZnO/p-CH_3_NH_3_PbI_3_/p-NiO_x_* solar cell with perovskite layer thickness of 200 nm (**a**), 400 nm (**b**), 600 nm (**c**), and 800 nm (**d**) on ETL and HTL layer thicknesses.

**Figure 5 nanomaterials-12-02692-f005:**
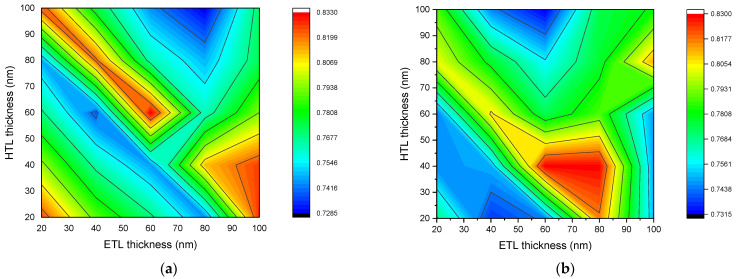
Dependence of fill factor of *n-ZnO/p-CH_3_NH_3_PbI_3_/p-NiO_x_* solar cell with perovskite layer thickness of 200 nm (**a**), 400 nm (**b**), 600 nm (**c**), and 800 nm (**d**) on the ETL and HTL layer thicknesses.

**Figure 6 nanomaterials-12-02692-f006:**
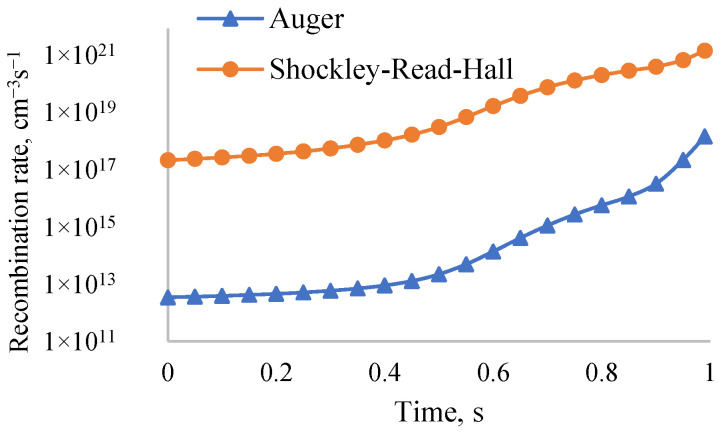
Dependence of SRH and Auger recombination rates of solar cell (200 nm, 60 nm, and 20 nm) on time.

**Figure 7 nanomaterials-12-02692-f007:**
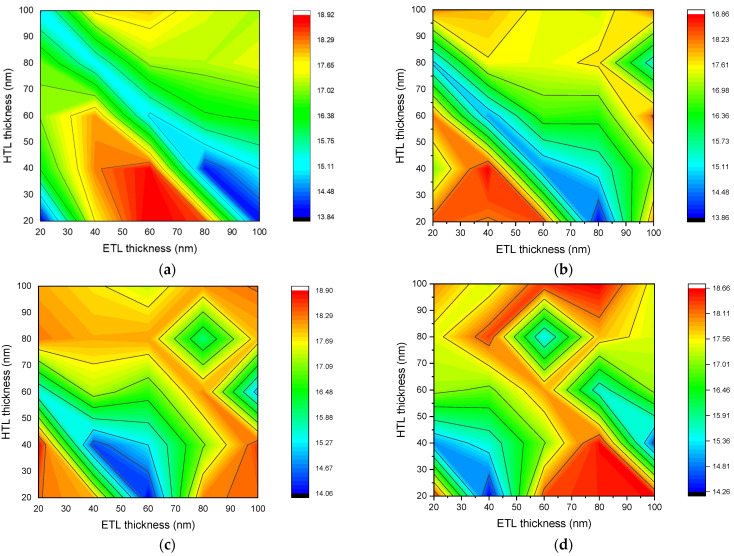
Dependence of output power of *n-ZnO/p-CH_3_NH_3_PbI_3_/p-NiO_x_* solar cell with perovskite layer thickness of 200 nm (**a**), 400 nm (**b**), 600 nm (**c**), and 800 nm (**d**) on ETL and HTL layer thicknesses.

**Figure 8 nanomaterials-12-02692-f008:**
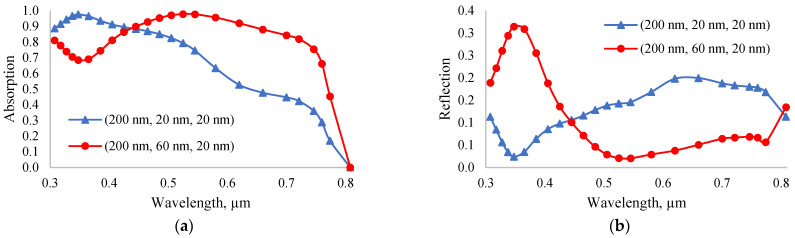
Dependence of the absorption (**a**) and reflection (**b**) coefficients of the solar cells with the lowest (200 nm, 20 nm, and 20 nm) and highest (200 nm, 60 nm, and 20 nm) output power on the light wavelength.

**Figure 9 nanomaterials-12-02692-f009:**
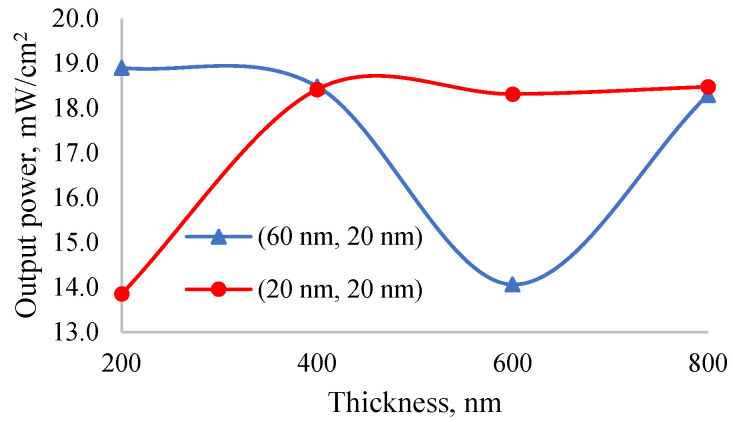
Dependence of output power of solar cell with ETL and HTL layer thicknesses of (60 nm, and 20 nm) and (20 nm, and 20 nm) on perovskite layer thickness.

**Table 1 nanomaterials-12-02692-t001:** Physical parameters of ZnO [30,31,32,33,34], CH_3_NH_3_PbI_3_ [35,36,37,38,39], and NiO_x._[40].

Parameters	ZnO (Al Doped)	CH_3_NH_3_PbI_3_	NiO_x_
E_g_, eV	3.3	1.56	3.8
χ, eV	4.5	3.9	1.6
N_c_, sm^−3^	2.2 × 10^18^	2.2 × 10^18^	1 × 10^18^
N_v_, sm^−3^	1.8 × 10^19^	1.8 × 10^19^	1 × 10^18^
μ_e_, sm^2^/Vs	50	2.5	2.8
μ_h_, sm^2^/Vs	5	5	2.8
ε	8.5	6.5	11
τ_we_, ps	0.84	0.33	-
τ_wh_, ps	0.15	1.87	-
I_e_, eV	0.054	0.08	0.3

## Data Availability

Data will be made available upon request from corresponding author.

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
