# Peer review of "Geometric Optimization of Perovskite Solar Cells with Metal Oxide Charge Transport Layers"

_nanomaterials, 2022, doi:10.3390/nano12152692_

Round 1

Reviewer 1 Report

In this paper, the authors provide a detailed investigation on the thickness’ effect of the various layers of n-ZnO/p-CH3NH3PbI3/p-NiOx perovskite solar cells on device parameters such as the photocurrent density, the open-circuit voltage, the produced power, the absorption/reflection etc. through optical simulations. Simulation results show that the maximum efficiency is obtained using a perovskite layer thickness of 200 nm, an electron transport layer (ETL) thickness of 60 nm, and a hole transport layer (HTL) thickness of 20 nm. The main result is that the maximum value of each of the photoelectric parameters is found in structures of different thicknesses for ETL. HTL and the photoactive perovskite layer.

By comparing with the state-of-the-art literature and the motivation that exists for optical modelling of perovskite solar cells, this work meets the requirements for consideration for publication in Nanomaterials. It highlights the importance of optical simulations of perovskite solar cells aiming at further improvement of their performance and is expected to be of interest for reasearchers in this field. However, some points in the manuscript should be furhter clarified and, for that reason, specific comments are provided below that need to be carefully addressed in order to improve the quality of the manuscript. 1. Genratlly, optical modelling of multilayer devices such as solar cells employs the materials refractive indices and the various layers' thicknessses ; however, thin films of metal oxides as well as of the perovskite have optical characteristics that are generally affected by the method of preparation and film deposition (this is especially true for metal oxides such as ZnO and NiO employed in the device structure that is optically simulated in this work).
Have the authors taken into account this effect when using data from the literature for all the employed solar cell layers ? Moreover, film roughness especially if the RMS value is in the range of some nanometers or more might also need to be considered even for planar perovskite solar cells (one way to model this is Bruggeman effective medium approximation).
Could the authors comment on this important issue ?
2. A plot of the distribution of the optical electric field intensity inside the various layers of the modeled perovskite solar cells versus wavelength for a wide spectral range (eg. 400-800 nm) would be really helpful to better understand how light is distributed for different wavelengths. This is important as, for example, increased field intensity does not necessarilty translate to an increased absorbance. 3. Some sentences are too long to understand their meaning for readers and, thus, tt's better to modify them in shorter sentences. 4. There are a few typos and grammar errors throughout the manuscript, which needs to be corrected. The authors should check the whole manuscript carefully to correct all such errors.

Author Response

In this paper, the authors provide a detailed investigation on the thickness’ effect of the various layers of n-ZnO/p-CH3NH3PbI3/p-NiOx perovskite solar cells on device parameters such as the photocurrent density, the open-circuit voltage, the produced power, the absorption/reflection etc. through optical simulations. Simulation results show that the maximum efficiency is obtained using a perovskite layer thickness of 200 nm, an electron transport layer (ETL) thickness of 60 nm, and a hole transport layer (HTL) thickness of 20 nm. The main result is that the maximum value of each of the photoelectric parameters is found in structures of different thicknesses for ETL. HTL and the photoactive perovskite layer.

By comparing with the state-of-the-art literature and the motivation that exists for optical modelling of perovskite solar cells, this work meets the requirements for consideration for publication in Nanomaterials. It highlights the importance of optical simulations of perovskite solar cells aiming at further improvement of their performance and is expected to be of interest for researchers in this field. However, some points in the manuscript should be further clarified and, for that reason, specific comments are provided below that need to be carefully addressed in order to improve the quality of the manuscript.

  1. Generally, optical modelling of multilayer devices such as solar cells employs the materials refractive indices and the various layers' thicknesses ; however, thin films of metal oxides as well as of the perovskite have optical characteristics that are generally affected by the method of preparation and film deposition (this is especially true for metal oxides such as ZnO and NiO employed in the device structure that is optically simulated in this work).
    Have the authors taken into account this effect when using data from the literature for all the employed solar cell layers ? Moreover, film roughness especially if the RMS value is in the range of some nanometers or more might also need to be considered even for planar perovskite solar cells (one way to model this is Bruggeman effective medium approximation).
    Could the authors comment on this important issue?

Answer: You are right. The refractive indices of metal oxides strongly depend on the synthesis method. In collecting the properties of materials from experimental studies, we paid attention synthesis method of materials.  We added some information to article: “Properties of materials were chosen according to synthesis of materials. In the future, to conduct the experiment based on simulation result we employed materials which were synthesized by using Sol-Gel method. Because, it is possible to grow this structure by using Sol-Gel method.” In this article, Transfer Matrix Method (TMM) is used for optical simulation. It is possible to consider scattering at rough interface of planar multilayer structure by using TMM in Sentaurus Device. So, we added some information to article about simulation of scattering at rough surface: “TMM can be used to model the effect of light trapping owing to scattering of incident light rough interface of a planar multilayer structure. Part of incident light at a rough interface is scattered and spreads incoherently.  In Sentaurus Device, TMM method use Scalar scattering theory to approximate the amount of scattered light at a rough interface. Besides, directional dependency of the scattering process is modeled by Angular Distribution functions (ADF). The so-called haze parameter defines the ratio of diffused (scattered) light to total (diffused + specular) light. In this scenario, a rough interface is characterized by its haze function (haze parameter as a function of the wavelength of incident light) for reflection and transmission as well as the ADF for direct coherent light incident at the interface and the ADF for scattered light incident at the interface.”

  1. A plot of the distribution of the optical electric field intensity inside the various layers of the modeled perovskite solar cells versus wavelength for a wide spectral range (eg. 400-800 nm) would be really helpful to better understand how light is distributed for different wavelengths. This is important as, for example, increased field intensity does not necessarily translate to an increased absorbance.

Answer: You are right, plot of the optical electric field intensity inside the various layers is good to prove and analyze the results obtained in simulation. But, in this article, we focused on determining the main photoelectric parameters of solar cell not optic parameters to find the best structure with highest output power and photoelectric parameters. Only we use give the dependence of reflection and absorption coefficients of structures, which have maximum and minimum output power, in order to explain difference between photoelectric parameters of structures. We thought it is enough to give the absorption and reflection spectra.

  1. Some sentences are too long to understand their meaning for readers and, thus, it's better to modify them in shorter sentences.

Answer: We corrected them.

  1. There are a few typos and grammar errors throughout the manuscript, which needs to be corrected. The authors should check the whole manuscript carefully to correct all such errors.

Answer: All typos and grammar errors were corrected carefully.

All changes were marked with yellow. 

Reviewer 2 Report

The authors present a work in which they apply calculations for the predictions of optimized geometry in PSC development. Although the idea is nice, the paper is not sufficiently uptodate in the overall results. First, the use the oldest type of PVK that was already supplanted by inorganic multication pvk, for the moisture problems (already evidenced in the introduction). Second, the fact that improving the thickness of the pvk layer is not good for recombination is a basic concept that does not require simulation, as well as the resulting thickness layer are already values that researchers well know when preparing solar cells. Probably, the only part that is really intriguing is the doping of ETL with nano-objects, and I suggest the authors to concentrate only on this aspect in revising the paper.

Author Response

Answer: Thank you for your suggestions. You are right, it is not very important topic now when we study this structure as only solar cell. Optimal thickness had been almost studied for various solar cells. But, our goals were not directed only to ZnO/CH3NH3PbI3/NiOx solar cells. we know that in solar cell, efficiency and output power are significant parameters. So, thicknesses, which output power reaches maximum value, are determined in a lot of researches. We can look at ZnO/CH3NH3PbI3/NiOx structure as other optoelectronic device. In that case, determining the various optimal thicknesses, which each photoelectric parameter reaches maximum value, is important. Therefore, in this work, we studied and found optimal thicknesses of this structures to gain maximum short circuit current, open circuit voltage, fill factor and output power. Their significances were described in conclusion. We planned our next researches depending on this work. Because, determining of the optimal thickness of above structure is important our experimental works and our future works. About it, we gave some points in conclusion. We thought that it is better to study the effect of nanoobjects, nanotextures, metal nanoparticles and quantum dots on solar cell with optimal thickness than unknown thicknesses. Your suggested topic was planned for our next works. Besides, we added some information to conclusion, method and results sections to show and explain the significance of this work according to your suggestion. 

Reviewer 3 Report

In this article, the authors investigated the thickness’ effect of each layer on the photo-electric parameters of n-ZnO/p-CH3NH3PbI3/p-NiOx solar cells through various simulations. It has various flaws and needed to be resolved.

Can the authors provide some performance of emerging photovoltaics in the introduction to cover the superiority of the proposed study in the PV field?

There should be reasons as well for choosing NiOx and ZnO, besides those mentioned in the introduction section (page 2). ZnO is not stable under UV irradiation and stability is also one of the critical challenges to tackle. There must be some explanations and comparisons of other ETLs to justify the choice of specific ETLs for this study.

VOC does not only depend on the thickness of the layer. So, there is a need to explain this.

In line#276, the statement, “the fill factor of 276 the solar cell at thicknesses of (600 nm, 80 nm, 80 nm) is 83.35% for all structures” is quite confusing. Please explain. How the FF is related to parasitic resistances?

The authors optimized the thickness for effectively operating solar cells under outdoor illumination. If the light intensity drastically reduces, for instance in the case of indoor light sources, how it will affect the optimization parameters? For understanding this phenomenon, I suggest the authors include the important work (https://doi.org/10.1016/j.jpowsour.2021.230782; https://doi.org/10.1016/j.jmrt.2021.12.086; https://doi.org/10.1016/j.dyepig.2021.109624).

Please improve the axis and legends of the simulation figures because these are too small to read. Also, the other figures can be improved and drawn in more representable ways. Please check the legends of Figure 8. It will be better to replace 1 and 2 with proper terminologies.

I suggest revising the conclusion and making it concise. Most of the details covered in the conclusion are already discussed in the previous section.

Author Response

In this article, the authors investigated the thickness’ effect of each layer on the photo-electric parameters of n-ZnO/p-CH3NH3PbI3/p-NiOx solar cells through various simulations. It has various flaws and needed to be resolved.

  1. Can the authors provide some performance of emerging photovoltaics in the introduction to cover the superiority of the proposed study in the PV field?

Answer: At the end of the introduction, we added some information: “Determining the optimal thicknesses of each layer of perovskite solar cells will allow to create solar cell with high efficiency in experiment. Besides it opens the way to upgrade forthcoming researches because it is better to research on the effect of metal nanoparticles or textures on solar cell with optimal thickness.”

  1. There should be reasons as well for choosing NiOx and ZnO, besides those mentioned in the introduction section (page 2). ZnO is not stable under UV irradiation and stability is also one of the critical challenges to tackle. There must be some explanations and comparisons of other ETLs to justify the choice of specific ETLs for this study.

Answer: Yes, you are right. ZnO is not stable under UV radiation. But in this work we used from AL doped ZnO.  In Mario Prosa’s work (https://doi.org/10.1021/acsami.5b08255), this problem is discussed very well. So, we added some explanation to the article why we chose the Al doped ZnO as ETL layer: “There are some disadvantages of ZnO in using solar cell applications. Main of them is degradation under UV light. This problem is well discussed and solved in Mario Prosa’s work [23], it was found that enhancing the ultraviolet stability of solar cells by using Al doped ZnO depending on Al concentration and proved theoretically. Therefore, we used from Al doped ZnO in this work. Besides, Al doped ZnO is the best choice for perovskite solar cells as ETL layer because of its high conductivity, availability at low-cost and high electron mobility.”

  1. VOCdoes not only depend on the thickness of the layer. So, there is a need to explain this.

Answer: It was mistake. We deleted it from article.

  1. In line#276, the statement, “the fill factor of 276 the solar cell at thicknesses of (600 nm, 80 nm, 80 nm) is 83.35% for all structures” is quite confusing. Please explain. How the FF is related to parasitic resistances?

Answer: There were some mistakes. We changed the sentence: “The fill factor of the solar cell at thicknesses of (600 nm, 80 nm, 80 nm) is 83.35%. It is highest value of fill factor.” Parasitic resistance mainly effects on voltage and current at the maximum power point. You can see from this link: https://www.pveducation.org/pvcdrom/solar-cell-operation/effect-of-parasitic-resistances

  1. The authors optimized the thickness for effectively operating solar cells under outdoor illumination. If the light intensity drastically reduces, for instance in the case of indoor light sources, how it will affect the optimization parameters? For understanding this phenomenon, I suggest the authors include the important work (https://doi.org/10.1016/j.jpowsour.2021.230782; https://doi.org/10.1016/j.jmrt.2021.12.086; https://doi.org/10.1016/j.dyepig.2021.109624).

Answer: Thank you for your suggestions. We used references that you suggested to explain effect of indoor illumination.

  1. Please improve the axis and legends of the simulation figures because these are too small to read. Also, the other figures can be improved and drawn in more representable ways. Please check the legends of Figure 8. It will be better to replace 1 and 2 with proper terminologies.

Answer: Axis and legends of the figures has been improved. Legends of figure 9 and 8 is changed to better one.

  1. I suggest revising the conclusion and making it concise. Most of the details covered in the conclusion are already discussed in the previous section.

Answer: You are right. We repeated some information in conclusion. According to your suggestion, we deleted them and we added other information to enrich the conclusion.

All corrections were marked with yellow.

Round 2

Reviewer 1 Report

The reviewer comments have been properly addressed by the authors so the revised manuscript can be published as it is.

Reviewer 2 Report

The motivation for the work remains poor, in my opinion. However, the work now can be published.

Reviewer 3 Report

Please accept it. Thank you